# Impact of Community Palliative Care on Quality of Life among Cancer Patients in Bangladesh

**DOI:** 10.3390/ijerph20156443

**Published:** 2023-07-25

**Authors:** Mostofa Kamal Chowdhury, Shafiquejjaman Saikot, Nadia Farheen, Nezamuddin Ahmad, Sarwar Alam, Stephen R. Connor

**Affiliations:** 1Department of Palliative Medicine, Bangabandhu Sheikh Mujib Medical University (BSMMU), Dhaka 1000, Bangladesh; mkcadil@bsmmu.edu.bd (M.K.C.); farheen.cmc@gmail.com (N.F.); nezamcpc@gmail.com (N.A.); 2Compassionate Narayanganj (Community-Based Palliative Care Project), c/o Department of Palliative Medicine, Bangabandhu Sheikh Mujib Medical University (BSMMU), Dhaka 1000, Bangladesh; sjsaikot@gmail.com; 3Department of Clinical Oncology, Bangabandhu Sheikh Mujib Medical University (BSMMU), Dhaka 1000, Bangladesh; drsarwarbd@yahoo.com; 4Worldwide Hospice Palliative Care Alliance, London WC1X 9JG, UK

**Keywords:** compassionate community, Compassionate Narayanganj, community palliative care, standard oncological care, quality of life

## Abstract

Cancer, a leading cause of mortality worldwide, is often diagnosed at late stages in low- and middle-income countries, resulting in preventable suffering. When added to standard oncological care, palliative care may improve the quality of life (QOL) of these patients. A longitudinal observational study was conducted from January 2020 to December 2021. Thirty-nine cancer patients were enrolled in the Compassionate Narayanganj community palliative care group (NPC), where they received comprehensive palliative care in addition to oncological care. Thirty-one patients from the Dept. of Oncology (DO) at BSMMU received standard oncological care. In contrast to the DO group, the NPC group had a higher percentage of female patients, was older, and had slightly higher levels of education. At 10 to 14 weeks follow-up, a significant improvement in overall QOL was observed in the NPC group (*p* = 0.007), as well as in the psychological (*p* = 0.003), social (*p* = 0.002), and environmental domains (*p* = 0.15). Among the secondary outcomes, the palliative care group had reduced disability and neuropathic pain scores. Additionally, there were statistically significant reductions in pain, drowsiness, and shortness of breath, as well as an improvement in general wellbeing, based on the results of the Edmonton Symptom Assessment Scale—Revised. At the community level in Bangladesh, increased access to palliative care may improve cancer patient outcomes such as QOL and symptom burden.

## 1. Introduction

Every year, nearly 602,000 patients in Bangladesh need supportive and palliative care [1]. Palliative care is not yet integrated into the primary care system in Bangladesh, nor is there a component of community engagement. Palliative care aims to improve quality of life and alleviate suffering in seriously ill patients and their families. Since its inception in 2007, Bangabandhu Sheikh Mujib Medical University (BSMMU) has been providing palliative care through the Centre for Palliative Care, which eventually evolved into an autonomous Department of Palliative Medicine. At present, most patients receiving palliative care have a cancer diagnosis.

Cancer is still a serious public health issue across the world. Cancer is anticipated to affect 16.3 million people by 2040, with the majority living in low- and middle-income countries [2]. Most malignancies are diagnosed in these regions at advanced stages, when treatment choices are restricted or unavailable [3]. Cancer symptoms and treatment have a poor impact on patients’ quality of life (QOL) due to physical discomfort, emotional stress, and financial strain [4].

There were 156,775 new cancer diagnoses in Bangladesh in 2020 [5], and the number of cancer patients needing palliative care before and near the end of life exceeded 170,000 [1]. Due to insufficient diagnostic capacity, ignorance, and cost of treatment, only one-third of the cancer patients can access primary care, treatment, and regular follow-up facilities [6]. Cancer patients often continue treatments that no longer benefit their health status, instead of aligning treatment strategies to improve their QOL. An effective palliative care strategy can provide appropriate support and symptom control for patients with advanced cancers [7].

### Background

Palliative care is compassionate care for people with serious illnesses throughout the life course and should be integrated throughout the healthcare system. When added to standard cancer care, palliative care has been shown to improve many outcomes for patients and their caregivers, including symptom burden, QOL, mood, prognostic understanding, end-of-life outcomes, resource utilization, and even (in some cases) length of survival.

A comprehensive review of studies from Africa and Asia found that older patients (>65 years) had superior QOL in several dimensions (psychological, existential, and practical) compared with younger patients, which was associated with good coping strategies and social support from family and friends [8].

Groups including the American Society of Clinical Oncology (ASCO), National Comprehensive Cancer Network, and European Society of Medical Oncology have issued clinical opinions recommending the use of palliative care earlier in the course of disease, based on evidence of benefit to patients or their caregivers, and there is no evidence that early palliative care increases patient costs or causes patient or caregiver harm [9,10].

In a randomized controlled trial by Temel et al. [11], individuals with metastatic NSCLC who received early palliative care had greater QOL, reduced rates of depression, less aggressive end-of-life care, and increased survival (median 11.6 vs. 8.9 months; *p* = 0.02). Similarly, patients who received early palliative care had significantly higher 1-year survival rates (63 percent vs. 48 percent; *p* = 0.038) in the ENABLE III study [12]. Additionally, 2095 Medicare patients receiving hospice care compared to 2260 patients receiving standard care, in a matched nested case–control study, had improved overall survival, and for six diagnostic groups survival improved by up to 2.7 months [13].

Narayanganj City Corporation (NCC), established in 2011, is one of the city corporations of Bangladesh. Its total area is 72.43 square kilometers, with an approximate population of about 2 million people distributed among 27 wards [14].

To improve access to essential, quality healthcare services for women, men, girls, and boys affected by serious chronic and life-limiting conditions, a project popularly known as “Compassionate Narayanganj” was implemented by the Department of Palliative Medicine at BSMMU, Dhaka, in collaboration with the Worldwide Hospice Palliative Care Alliance (WHPCA) in NCC located in central Bangladesh. This 39-month (1 April 2018–30 June 2021) project was supported through a Community Partnership grant from UK Aid Direct. The project strategically focused on the development of a compassionate, people-centered, and people-led community movement to deliver home-based PC to people living with and affected by life-limiting illnesses, in collaboration with healthcare providers and related stakeholders, with the goal of improving the QOL of the poorest, with a focus on women and girls, to progress toward universal health coverage (UHC).

One of the goals was to increase access to palliative care for oncology patients in Narayanganj by integrating palliative care services into the local health system, as well as to investigate the costs and patient-reported outcomes of palliative care in this low-income context. This has been the first study of its sort in Bangladesh and builds directly on our previous work in this setting.

This study offers fresh and critical data on the impact of palliative care on QOL and the prior costs of medical treatment in this environment, in addition to formally introducing palliative care into the practice of medicine in Bangladesh. This, in turn, is critical for influencing healthcare policy and building momentum for increased and coordinated local and international financing to develop and integrate palliative care services within local health systems, safeguarding this human right for everyone.

## 2. Methodology

### 2.1. Study Design

A prospective longitudinal observational study among two groups of cancer patients.

### 2.2. Aim of the Study

To observe the impact of palliative care on health-related quality of life and symptom management among cancer patients.

### 2.3. Study Setting

The “Compassionate Narayanganj” program developed and implemented a novel community home-based palliative care service with a hub in NCC and an emergency telephone hotline to address the community’s unmet palliative care needs. The program employed and trained nine palliative care assistants (PCAs) from the same community. After receiving 6 months of basic training, they offered basic care and support (including physical, psychological, and spiritual care), instructed and supported family members during home visits, and referred patients as needed. A small clinical team of a nurse, two physicians, and one part-time physiotherapist made regular home visits to address clinical care requirements to give ongoing help. Additionally, the program included sensitization and awareness campaigns and training for local volunteers and health professionals to grasp the concept of palliative care and its potential benefits.

### 2.4. Study Population

Thirty-nine consecutive cancer patients from the NCC receiving community-based comprehensive palliative home care along with standard oncological care were enrolled in the NPC group and were followed up. On the other hand, from cancer patients receiving conventional oncological therapy at BSMMU’s oncology department but without receiving compassionate palliative care, 31 were identified and tracked. Among them, those who chose palliative care after enrolling but before the study period’s conclusion were omitted from the study.

Palliative care involved symptom assessment/treatment as well as emotional and spiritual support for the patient and family, as well as creating care objectives and supporting decision-making. All patients received standard oncological care throughout the study period, unless they or their families decided that standard oncological therapy no longer aligned with their care goals.

#### 2.4.1. Inclusion Criteria

Patients with any diagnosed cancer (other than non-malignant skin cancers).Ambulatory, able to respond verbally to questions, and intact cognitive function.

#### 2.4.2. Exclusion Criteria

Patients with non-cancer conditions as a primary diagnosis.Already receiving palliative care.Less than 18 years of age.

### 2.5. Data Confidentiality and Protection

Participants were assigned a numerical ID at recruitment. The copies of all patient records and all spreadsheets were kept locked at the main CPC office in Dhaka. All electronic data were password-protected. Only the research team had access to the data.

### 2.6. Data Sources and Instruments

To measure health-related quality of life, the WHOQOL-BREF (a shorter version of the WHOQOL-100) was used. With a total of 26 questions, it examines four essential areas: physical, psychological, social, and environmental domains. It is a self-administered survey in Bengali that has been validated for use in Bangladesh. The four domain ratings describe a person’s perception of their own quality of life in each category. The domain ratings are weighted in a positive direction (i.e., higher scores denote higher quality of life). The means and standard deviations of the domain scores were computed, and the means of the domain scores before and after palliative treatment were compared using the paired *t*-test [15].The Edmonton Symptom Assessment Scale—Revised (ESAS-R), a simple, validated, and reliable multi-item instrument designed to evaluate a variety of symptoms in palliative care patients, was used to assess symptom burden, which included both physical and psychological symptoms [16,17]. The ESAS-R has been demonstrated to be an effective audit tool for examining patterns of palliative symptom management and allowing for institutional comparisons [18,19]. A higher score for a symptom suggests greater intensity.The Washington Group (WG) Questions and the Palliative Performance Scale (PPS) were used to measure disability levels. The Washington Group on Disability Statistics designed, evaluated, and adopted a set of six questions on functioning for use in national censuses and surveys. The questions reflect improvements in the notion of disability and are based on the World Health Organization’s International Classification of Functioning, Disability, and Health [20]. The PPS employs five observer-rated dimensions that are associated with the Karnofsky Performance Scale (100–0). For cancer patients in outpatient and ambulatory settings, the PPS is a trustworthy and valid instrument that correlates well with actual survival and median survival time. It has been proven to be beneficial for detecting and tracking potential palliative care patients’ care needs, especially when these needs alter with the progression of illness [21].Neuropathic pain was assessed by the Douleur Neuropathique 4 (DN4) screening test. The DN4 screening test is a brief 10-item questionnaire that can be completed in <5 min. Patients with a score of ≥4 have a 90% chance of having a diagnosis of neuropathic pain (NeP). It should also be noted that the probability that the patient has a diagnosis of NeP is >90% if the DN4 score is 5 (93%) or 6 (98.5%) [22].

The ESAS-R, WG, PPS, and DN4 questionnaires were translated and linguistically validated as suggested by Oxford University Innovation [23] before being used for data collection. For this purpose, we conducted forward and back translation, review by an expert team and, finally, a pilot study involving patients. With feedback from patients and reviewers, necessary modifications were carried out, retested, and finalized. 

Direct and indirect costs (for medications, travel, treatments, hospitalization, ER use, clinic visits, etc.) were assessed verbally. We also assessed whether the patient or family sold property, took loans, or received gifts from relatives to pay for medical expenses. 

### 2.7. Data Collection

After providing written informed consent, the patients completed the baseline questionnaires at the time of enrollment and at 10 ± 4 weeks from baseline. The research coordinator, bilingual in Bengali and English, gathered data on the questionnaires, recorded the patients’ responses, and clarified for incomplete answers. 

### 2.8. Analysis Plan

The primary outcome was the difference in WHOQOL-BREF score from baseline to follow-up visit at 6 to 14 weeks. With 100 patients, we predicted that the trial would have 80% power to detect a significant intergroup difference in the change in instrument scores from baseline to 10 ± 4 weeks, with a mean effect size of 0.5 SD. The secondary outcomes were differences in ESAS-R, Palliative Performance Scale, and Washington Group Question scores across the two arms from baseline through follow-up. SPSS (Version 21.0) software was used for statistical analysis. Differences between study arms were analyzed using *t*-tests for continuous variables and chi-squared tests for categorical variables at the baseline and follow-up visits.

## 3. Results

In the Compassionate Narayanganj palliative care oncology group (NPC) and the BSMMU Department of Oncology group (DO), 39 and 31 patients were enrolled, respectively. In the NPC and oncology groups, the average age was 49.15 ± 11.91 and 39.87 ± 14.68 years, respectively. In the NPC group, 74.4 percent were female, whereas 71% were male in the oncology group. The vast majority of patients were Muslim and married. In both groups, the primary caregiver was the spouse (Table 1).

Breast and cervical cancers were more common in the NPC group, but colorectal and lung cancers were more common in the oncology group. The majority of Narayanganj patients were at stages 2 and 3, as opposed to stages 1 and 2 in the comparison group. In both groups, more than 90% of patients were aware of their diagnosis, which was revealed by clinicians in the majority of cases. In Narayanganj, half of the patients recognized their prognosis, while in the oncology group more than 70% did not. Hypertension and diabetes were frequent in the NPC group, although comorbidities were less common in the other group (Table 2). Patients in both groups received surgery, chemotherapy, and radiation. Biological, hormonal, and supplementary medicines were uncommon.

Comprehensive palliative care demonstrated a considerable increase in overall quality of life in Narayanganj after 6 to 14 weeks of follow-up. Only the realm of social connections improved in the oncology group (0.028). However, palliative care resulted in statistically significant improvements in the overall quality of life (0.007), psychological health (0.003), social relationships (0.002), and environmental health (0.015) domains (Table 3).

Palliative treatment resulted in considerable improvements in pain, drowsiness, shortness of breath, and general wellbeing (Table 4). Anti-emetics and PPIs were the most regularly utilized medications. Higher usage of opioids, adjuvants, and NSAIDs resulted in improved pain management in the palliative care group. The number of patients with disability increased by 33.3% in the oncology group but only 10.52% in the palliative care group (Table 5). The average palliative performance scale value decreased by 9.3 percent among palliative care recipients, compared to a 14.18 percent decrease among conventional oncology treatment recipients. Furthermore, neuropathic pain grew less in the NPC group than in the oncology group (2.5% vs. 19.4%).

Although the mean monthly income and direct treatment-related costs were higher in the oncology group, total treatment expenses were very similar between the two groups (Table 6). Inpatient admission was greater in the oncology arm (96.8% vs. 51.4%), although emergency visits were higher in the palliative care arm (20.5% vs. 3.2%). Common transportation options included three-wheelers, rickshaws, and buses.

## 4. Discussion

The importance of this paper is that there is very little literature on the impact of community-based palliative care programs in limited-resource settings. Almost all of the literature is based on findings from high-income countries.

Patients with advanced cancer frequently encounter symptoms due to illness and treatment that add to their anguish and reduce their quality of life (QOL). Care targeted at controlling these symptoms, whether the patient is receiving continuing disease-directed medication to manage the malignancy or not, is thus an important component of high-quality patient-centered care [24].

Palliative care in Bangladesh is still in its infancy, with cancer patients accounting for a significant portion of patients receiving care. Similarly, the Australian Health Ministry reported that over 80% of admissions to palliative care units were for malignancies [25].

Most subjects in our study, especially in the NPC group, were female, with mean ages of around 50 and 40 years in the NPC and oncology groups, respectively. Breast (25.6%) and cervical (15.4%) cancers were more prevalent in the NPC group, but colorectal (16.1%) and lung cancers (16.1%) were more common in the oncology group. In cross-sectional research conducted at a Malaysian palliative care facility, 53 percent of patients were female, with a mean age of 54 years. Breast cancer (30%) and lower gastrointestinal cancer (24%) were the two most common reasons for admission to a palliative care unit, followed by malignancies of the upper gastrointestinal tract (18%), hepatobiliary system (15%), pancreatic malignancies (7%), and other soft-tissue tumors (6%) [26].

In our study, palliative care significantly improved the overall QOL among cancer patients, which was corroborated by a decrease in symptoms as measured by the ESAS-R scores. A 5-year randomized controlled trial on 322 patients newly diagnosed with advanced cancer in Lebanon found that those who received palliative care treatments in addition to oncological therapy had greater QOL and mood ratings than those who only received oncological care [27]. Early referral to palliative care not only allows for quick diagnosis and treatment of symptoms but also reduces caregiver stress and aggressive measures near the end of life [28]. Evidence indicated that after 3 and 6 months of follow-up, the QOL among early palliative care recipients was greater on average. Furthermore, patients had decreased symptom severity and a longer survival time [29].

Palliative treatment was linked with statistically significant and clinically meaningful improvements in QOL at 1 to 3 months in a meta-analysis of 15 studies examining QOL at the 1–3-month follow-up [30]. Even if the palliative care team is consulted late in the patient’s disease trajectory, a beneficial effect on the QOL of hospitalized patients with advanced cancer can be seen [31].

Palliative care is a holistic approach to treatment that aims to improve the QOL of patients with advanced chronic conditions or in end-of-life circumstances by alleviating physical and non-physical suffering and ensuring a dignified death. At the time of follow-up, the palliative care recipients had significantly improved quality in the psychological, social, and environmental domains, according to our observations. In 2010, a WHOQOL-BREF questionnaire survey of 60 patients at Chennai’s Jeevodaya Hospice Treatment Centre revealed an increase in QOL in both the psychological and environmental domains following 15 days of palliative care [15]. Because patients are suffering from terminal illnesses, improvement in the functional domain is highly improbable in palliative care, even if symptoms are controlled.

It is hard to alleviate all symptoms of terminally ill patients, but based on the findings of this study we were able to significantly reduce long-term symptoms such as pain, sleepiness, and shortness of breath, and to improve overall wellbeing. This differs from the findings of the Malaysian surgical palliative care facility, where not only pain, nutritional deficiency, and inadequate tissue oxygenation were eased, but also emergency life-threatening problems such as obstruction, bleeding, and life-threatening infections [26]. This disparity might be attributable to a lack of surgical facilities in the community setting.

In a study of 162 patients measured by ESAS-R, the mean scores decreased continuously during hospitalization (ANOVA for repeated measures: *p* < 0.0001). All symptoms experienced a statistically significant decrease in intensity between day 1 and day 7 among individuals with moderate/severe symptoms at baseline [32]. Again, in a large cluster-randomized controlled study of patients with advanced solid tumors, those who received early palliative care experienced a substantial reduction in symptom intensity according to the ESAS-R at 4 months when compared to those who received standard treatment [33].

Among the two groups, average monthly income was higher in the oncology group, but total treatment expense was similar. The cost of current ongoing medications among the palliative treatment receivers before enrollment was almost double. Our study was limited in that we did not track the expenses of therapy and their impact over the course of the investigation, but during the yearly survey for monitoring of the project on query it was evident that the NPC participants had lower hospitalization rates than before enrollment in this project. Community-based palliative care programs are associated with fewer hospitalizations and lower expenses in the final months of life [34,35]. An early focus on treatment aimed at increasing QOL has been shown to enhance patient satisfaction, reduce sadness and anxiety, improve resource utilization, improve survival, and reduce total healthcare expenditure [12,13,36]. Palliative care consultation, when begun within two days of admission for patients with cancer and many comorbidities, resulted in a 32% reduction in hospital expenses, according to May et al. [37].

Even after adjusting for pain severity, studies demonstrate that people with neuropathic pain have higher pain intensity, a lower QOL, and a larger impact on daily functioning than those with nociceptive (or inflammatory) pain [38,39,40]. According to preliminary research, up to 39% of cancer patients may develop neuropathic pain [41,42].

At baseline, 38.5 percent of NPC patients experienced neuropathic pain, compared to 12.9 percent of oncology care recipients. However, neuropathic pain emerged more frequently in the second group at the follow-up. The palliative care group’s likely increased usage of opioids, analgesics, and neuropathic adjuvants resulted in decreased neuropathic pain development over time. In contrast, 17 percent of patients in research conducted across 17 European centers had neuropathic pain. Patients with neuropathic cancer pain were more likely to be receiving oncological therapy, powerful opioids, and adjuvant analgesia, as well as having a worse performance status [38].

Home-based treatment is especially crucial, since hospitals can hasten functional loss in patients with terminal disease [43]. However, there were no changes in QOL identified in people referred to a community palliative care program [44], and a study of patients with lung cancer indicated that those with good QOL did not change as they approached the end of their life [45]. In contrast, our community-based palliative home care program was successful in enhancing QOL.

Most patients in both groups were aware of their diagnosis, revealed by clinicians in almost all cases. Half of the palliative patients recognized their prognosis, while in the oncology group more than 70% did not. Unfortunately, many patients with advanced, incurable illness have misunderstandings about their prognoses and the aims of their cancer therapy. Weeks et al. [46] discovered that 69 percent of lung cancer patients and 81 percent of colorectal cancer patients incorrectly believed that their palliative chemotherapy regimens were provided with the purpose to cure [46]. Surprisingly, early palliative care alleviates this issue. Temel et al. found that patients with advanced NSCLC who were randomly allocated to early palliative care were more likely to maintain or gain an accurate awareness of their prognosis than patients who received conventional treatment (82.5% vs. 59.6%; *p* = 0.02) [47].

When faced with a serious, life-limiting disease, people with physical disabilities deserve compassionate, high-quality, and effective palliative and end-of-life care. Patients with cognitive impairments may not obtain adequate-quality palliative care during their journey with a life-threatening illness such as HIV/AIDS or cancer, owing to a knowledge gap and a lack of skills among healthcare professionals [48]. In our research, in the oncology group, the number of patients with impairment increased by 33.3 percent, but only by 10.52 percent in the palliative care group. The Palliative Performance Scale also showed less decline towards the end of life in the palliative care group. Comprehensive palliative care proved effective in reducing disability and performance deterioration.

## 5. Limitations

This study has several limitations. First, there were significant differences in the age, sex, cancer site, and staging of the participants in the oncology versus palliative care groups that could have influenced the results. This calls into question the comparability of the oncology control group to the palliative care group. Second, because of COVID-19, healthcare services were restricted, and data collection was much more difficult, resulting in a limited sample size. Third, a financial estimate for cost-effectiveness is needed to support the expansion and sustainability of palliative care services in this and other low-income settings. Proving the economic worth of palliative care in LMICs is critical to long-term sustainability. The cost–benefit ratio could not be calculated, since the spending was not tracked. Fourth, more systematic monitoring and assessment from the start of the research would have resulted in more diverse data on the impact of palliative care on the study’s patients.

## 6. Conclusions

This research demonstrated that community palliative care can improve the quality of life of individuals with cancer. In Bangladesh, where palliative care is infrequently practiced, a community-based strategy based on this paradigm might play a key role in increasing access to palliative care. There is a problem in determining how to improve palliative care integration to improve patient outcomes. Increased engagement and understanding of palliative care, even at the community level, though not measured in this study, may improve cancer patient outcomes such as quality of life, symptom load, and perhaps even survival. Future studies with larger cohorts and qualitative research are needed to better understand the strengths, weaknesses, and obstacles to this community-based strategy. The lessons learned from this initiative will be used in the development of palliative care throughout Bangladesh.

## Figures and Tables

**Table 1 ijerph-20-06443-t001:** Demographic characteristics of both groups.

	NPC (*n* = 39)		Oncology (*n* = 31)	
	Frequency	Percent	Frequency	Percent
Age (mean ± SD)	49.15 ± 11.91		39.87 ± 14.68	
Median age	48.00		38.00	
Gender				
Male	10	25.6	22	71.0
Female	29	74.4	9	29.0
Religion				
Islam	35	89.7	29	93.5
Hinduism	4	10.3	2	6.5
Marital status				
Unmarried	1	2.6	6	19.4
Married	30	76.9	23	74.2
Widowed/widower	7	17.9	1	3.2
Separated	1	2.6	1	3.2
Average number of family members	4.9~5		6.16~6	
Main caregiver				
Spouse	16	41	19	61.3
Children	12	30.8	3	9.7
Parent	1	2.6	4	13
Others	10	25.6	5	16.1
Education:				
Can write name only	18	46.2	1	3.2
Up to graduation	15	38.5	20	64.5
Post-graduation/Kamil	1	2.6	4	12.9
No education	5	12.8	6	19.4
Occupation:				
Housewife	25	64.1	4	12.9
Farmer	-	-	8	25.8
Student	-	-	3	9.7
Unemployed	2	5.1	1	3.2
Others	12	30.8	15	48.4
Personal history: (multiple responses)				
None	22	56.4	18	58.1
Smoking and tobacco chewing	6	15.4	9	29
Betel leaf & betel nut	15	38.5	13	41.9
Others	3	7.7	3	9.7

**Table 2 ijerph-20-06443-t002:** Frequency of diagnosis and comorbidities.

	NPC (*n* = 39)		Oncology (*n* = 31)	
Malignancy (NPC)	Frequency	Percentage	Frequency	Percentage
Breast	10	25.6	2	6.5
Cervix	6	15.4	-	-
Colorectal	5	12.8	5	16.1
Lung	5	12.8	5	16.1
Ovary	2	5.1	2	6.5
Others	11	28.2	17	54.8
Stages of malignancy:				
Stage 1	6	15.4	10	32.3
Stage 2	12	30.8	11	35.5
Stage 3	15	38.5	6	19.4
Stage 4	6	15.4	4	12.9
Metastasis:	13	33.3	8	25.8
Lung	1	2.6	-	-
Lymph node	4	10.3	3	9.7
Liver	2	5.1	1	3.2
Bone	3	7.7	-	-
Disseminated	-	-	2	6.5
Others	3	7.7	2	6.4
Patient knows diagnosis:				
No	3	7.7	2	6.5
Yes	36	92.3	29	93.5
If yes, revealed by				
Doctor	34	94.4	28	96.6
Family members	1	2.8	1	3.4
Guess	1	2.8	0	0.0
Patient knows prognosis:				
No	19	48.7	22	71.0
Yes	20	51.3	9	29.0
Comorbidities: (multiple responses)				
None	17	43.6	26	83.9
Hypertension	9	23.1	2	6.4
Diabetes mellitus	7	17.9	2	6.4
Bronchial asthma	5	12.8	-	-
Chronic obstructive pulmonary disease (COPD)	5	12.8	-	-
Others	7	17.9	1	3.2

**Table 3 ijerph-20-06443-t003:** Comparison of changes in quality of life after 10 ± 4 weeks.

Paired-Samples Test									
		Paired Differences					t	df	Sig. (2-tailed)
		Mean	Std. Deviation	Std. Error Mean	95% Confidence Interval of the Difference				
					Lower	Upper			
Overall all domains	NPC	2.07387	4.55982	0.73016	0.59575	3.55199	2.840	38	0.007 *
	Oncology	−0.73886	5.41998	0.97346	−2.72693	1.24920	−0.759	30	0.454
Domain 1: Physical health	NPC	−0.16117	1.89951	0.30417	−0.77692	0.45458	−0.530	38	0.599
	Oncology	−0.14747	1.20739	0.21685	−0.59034	0.29541	−0.680	30	0.502
Domain 2: Psychological health	NPC	0.61538	1.18849	0.19031	0.23012	1.00065	3.234	38	0.003 *
	Oncology	0.45161	1.90428	0.34202	−0.24688	1.15011	1.320	30	0.197
Domain 3: Social relationships	NPC	1.19658	2.20143	0.35251	0.48296	1.91020	3.394	38	0.002 *
	Oncology	−0.81720	1.96796	0.35346	−1.53906	−0.09535	−2.312	30	0.028 *
Domain 4: Environmental health	NPC	0.42308	1.04213	0.16687	0.08526	0.76090	2.535	38	0.015 *
	Oncology	−0.22581	1.94010	0.34845	−0.93744	0.48583	−0.648	30	0.522

* = significant value.

**Table 4 ijerph-20-06443-t004:** Changes in ESAS-R analysis.

		Paired Differences					t	df	Sig. (2-tailed)
		Mean	Std. Deviation	Std. Error Mean	95% CI of the Difference				
					Lower	Upper			
Pain	NPC	−1.026	2.777	0.445	−1.926	−0.126	−2.307	38	0.027 *
	Oncology	0.516	2.488	0.447	−0.397	1.429	1.155	30	0.257
Tiredness	NPC	0.179	3.456	0.553	−0.941	1.300	0.324	38	0.747
	Oncology	0.677	2.358	0.423	−0.187	1.542	1.600	30	0.120
Drowsiness	NPC	1.000	3.078	0.493	0.002	1.998	2.029	38	0.050 *
	Oncology	0.065	1.731	0.311	−0.570	0.699	0.208	30	0.837
Nausea	NPC	0.333	2.994	0.479	−0.637	1.304	0.695	38	0.491
	Oncology	−0.323	1.904	0.342	−1.021	0.376	−0.943	30	0.353
Lack of appetite	NPC	0.256	3.747	0.600	−0.958	1.471	0.427	38	0.672
	Oncology	−0.677	2.166	0.389	−1.472	0.117	−1.741	30	0.092
Shortness of breath	NPC	−1.821	3.339	0.535	−2.903	−0.738	−3.405	38	0.002 *
	Oncology	0.355	2.702	0.485	−0.636	1.346	0.731	30	0.470
Depression	NPC	−1.077	3.437	0.550	−2.191	0.037	−1.957	38	0.058
	Oncology	0.065	0.680	0.122	−0.185	0.314	.528	30	0.601
Anxiety	NPC	−0.692	3.381	0.541	−1.788	0.404	−1.279	38	0.209
	Oncology	0.677	2.625	0.472	−0.286	1.640	1.437	30	0.161
Wellbeing	NPC	−1.513	2.761	0.442	−2.408	−0.618	−3.421	38	0.002 *
	Oncology	0.194	2.562	0.460	−0.746	1.133	0.421	30	0.677

Paired-samples test conducted; CI = confidence interval; * = significant value

**Table 5 ijerph-20-06443-t005:** Comparison of changes in Washington Group questionnaire analysis.

		Number of Patients with Disability	1 Domain Scored “a Lot of Difficulty” or “Unable to Do It”	2 Domains Scored “a Lot of Difficulty” or “Unable to Do It”	3 Domains Scored “a Lot of Difficulty” or “Unable to Do It”	4 Domains Scored “a Lot of Difficulty” or “Unable to Do It”	5 Domains Scored “a Lot of Difficulty” or “Unable to Do It”	6 Domains Scored “a Lot of Difficulty” or “Unable to Do It”
NPC (N = 39)	Initial	19 (48.72%)	10 (25.64%)	6 (15.38%)	3 (7.69%)	0	0	0
	F-up	21 (53.85%)	5 (12.82%)	6 (15.38%)	6 (15.38%)	4 (10.26%)	0	0
Oncology, BSMMU (N = 31)	Initial	3 (9.68%)	1 (3.23%)	2 (6.45%)	0	0	0	0
	F-up	4 (12.90%)	2 (6.45%)	1 (3.23%)	1 (3.23%)	0	0	0

**Table 6 ijerph-20-06443-t006:** Monthly mean income and mean cost (in BDT) analysis before enrollment (excluding outliers).

	NPC	Oncology
Monthly mean income	14,324.32	22,677.42
Treatment cost	89,405.41	123,741.94
Investigation cost	36,501.35	34,148.39
Transportation cost	77,027.03	69,193.55
Costs of current ongoing medications	13,439.15	6993.29
Other variable costs	46,418.92	51,750.00
Total treatment expense (according to patient/caregiver)	262,791.85	285,827.16

## Data Availability

The data presented in this study are available upon request from the lead author. The data are not publicly available due to the potential that personally protected health information could be used to identify study subjects.

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
