# Peer review of "Impact of Community Palliative Care on Quality of Life among Cancer Patients in Bangladesh"

_ijerph, 2023, doi:10.3390/ijerph20156443_

Round 1

Reviewer 1 Report

This is an important study of practical relevance. I recommend its publication; but have a few suggestions for improvement which I have indicated which track changes (attached). Most of them are minor; but I have two important points to highlight. The first is the statement that palliative care is specialised care which is objectionable because while it can be specialised, it need not be. As recommended by the World Health Assembly resolution 67 of 2014 and as evident by a growing number of publications, much of palliative care is to be practised by every healthcare provider, integrated into healthcare 'at all levels, across the continuum of care'.

Secondly, while much of the improvement in QOL could well be because of community engagement, that point has not been proved and therefore the relevant sentence has to be modified in the conclusions.

Reviewer 2 Report

As stated in the article the study has several limitations. Among them significant are:  1) limited sample size for a very ambitious set of questions and comparisons and 2) significant differences in the age, sex, cancer site and staging of the participants in the oncology versus palliative care group.

Reviewer 3 Report

This is an interesting report in an area where there is limited research - ie low and middle income countries

There are limitations in the small number of patients in both arms and the differences between the groups. This has been discussed in the Limitations section

There are other areas to consider changes:

Line 55-58 - This is needs to be referenced or clearer that this is an introduction to the following paragraphs

Line 266 onwards - it is unclear if the discussion is about this paper and the results the authors found or other papers . This needs to be clarified

Line 293 this reads as if well being was reduced and needs to be and improve overall well-being

Line 359 limitations The relatively small number of patients and the differences in the groups should be the first limitation discussed, as it is crucial to the results. this needs to be expanded as there are concerns about the groups

The English is variable and could benefit from a review

eg

line 16 got would be better as undertaken

line 36 Since its inception

line 556 improves may be better as has been shown to improve

line 66 and there is no 

line 289 progress may be better as improvement

line 354 In should be in
